# Benchmarking and Improving Text-to-SQL Generation under Ambiguity

**Adithya Bhaskar**[*♠◇]  **Tushar Tomar**[*♠]  **Ashutosh Sathe**[♠]  **Sunita Sarawagi**[♠]

♠IIT Bombay  ◇Princeton University

## Abstract

Research in Text-to-SQL conversion has been largely benchmarked against datasets where each text query corresponds to one correct SQL. However, natural language queries over real-life databases frequently involve significant ambiguity about the intended SQL due to overlapping schema names and multiple confusing relationship paths. To bridge this gap, we develop a novel benchmark called AmbiQT with over 3000 examples where each text is interpretable as two plausible SQLs due to lexical and/or structural ambiguity.

When faced with ambiguity, an ideal top-$k$ decoder should generate all valid interpretations for possible disambiguation by the user (Elgohary et al., 2021; Zhong et al., 2022). We evaluate several Text-to-SQL systems and decoding algorithms, including those employing state-of-the-art LLMs, and find them to be far from this ideal. The primary reason is that the prevalent beam search algorithm and its variants, treat SQL queries as a string and produce unhelpful token-level diversity in the top-$k$.

We propose LogicalBeam, a new decoding algorithm that navigates the SQL logic space using a blend of plan-based template generation and constrained infilling. Counterfactually generated plans diversify templates while in-filling with a beam-search, that branches solely on schema names, provides value diversity. LogicalBeam is up to $2.5\times$ more effective than state-of-the-art models at generating all candidate SQLs in the top-$k$ ranked outputs. It also enhances the top-5 Exact and Execution Match Accuracies on SPIDER and Kaggle DBQA[1].

## 1  Introduction

Research on Text-to-SQL generation has focused on scenarios where each natural language question is associated with one correct SQL (Zelle and Mooney, 1996; Tang and Mooney, 2000; Scholak et al., 2021a; Wang et al., 2020; Rubin and Berant, 2021; Xie et al., 2022; Arcadinho et al., 2022; Zeng et al., 2022; Scholak et al., 2021b; Pourreza and Rafiei, 2023). Popular benchmarks driving such research, including WikiSQL (Zhong et al., 2018), SPIDER (Yu et al., 2018), its robust perturbations (Chang et al., 2023), and even "in-the-wild" benchmarks such as KaggleDBQA (Lee et al., 2021) and SEDE (Hazoom et al., 2021) all associate one correct SQL with text. Meanwhile, ambiguity is prevalent in real-life databases — particularly the ones obtained by integrating several data sources for data analysis, where a natural language interface is most in demand. The sources of ambiguity are several — inherent ambiguity of natural language, the user's ignorance of table/column names, overlapping strings in column names, underspecified clauses, and confusion about whether aggregates are pre-computed, or if a join is required. Hazoom et al. (2021) observe that up to 87% of queries on the stack exchange database are underspecified, and Wang et al. (2022) mention that 11% of queries exhibited ambiguity in column names. Although prior work has brought up ambiguity, there is no publicly available benchmark with ambiguous queries, nor a comprehensive evaluation of systems under ambiguity.

Our first contribution is to bridge this gulf by developing a benchmark, AmbiQT, that tests *performance under ambiguity* in the context of current models. AmbiQT includes over 3000 examples, each associating a natural question on a database with *two* valid SQLs. Inspired by our experience with several real-world datasets, we target four types of ambiguity spanning both lexical (ambiguous column and table names) and structural (whether a join is necessary, and an aggregate is pre-computed) ambiguity. The benchmark is generated via a combination of ChatGPT (OpenAI, 2022) based synonym generation and perturbation,

---

[1]We release AmbiQT and LogicalBeam's implementation publicly at `https://github.com/testzer0/AmbiQT`.

* Equal Contribution. Work done while AB was at IIT Bombay. Correspondence to: <adithyabcse@gmail.com>

and standard rule-based perturbation.

When faced with ambiguity, an ideal Text-to-SQL system should incorporate all valid alternatives in their top-$k$ SQL outputs, for user resolution. We show that present approaches, ranging from T5-3B (Raffel et al., 2019) to SOTA models, fail to generate all ambiguous outputs with any decoding strategy, including beam search and diversity-promoting sampling methods such as Nucleus (Holtzman et al., 2020) and Typical sampling (Meister et al., 2023). Most outputs are small lexical tweaks of the top choice, bringing about little meaningful diversity in SQL structures or schema alternatives. Even SOTA LLMs like Chat-GPT (OpenAI, 2022) suffer from this issue.

To remedy the lack of diversity, we propose a new decoding algorithm, LogicalBeam, that allocates branching to explore underlying logical variants of the SQL rather than the string form. We catalog the errors of T5-3B (Raffel et al., 2019) on the SPIDER dev split and use our insights to encourage targeted types of diversity — the number of JOINs and selections, and table/column names.

Our main contributions are:

- We develop AmbiQT, the first benchmark that tests performance under four types of ambiguity over **3000+** examples.

- We show that SOTA methods, including a fine-tuned T5-3B, RESDSQL (Li et al., 2023), OpenAI Codex, and ChatGPT, provide a poor representation of ambiguity despite their high accuracy on conventional benchmarks.

- We present LogicalBeam, a two-step algorithm that generates plan-based templates with counterfactually controlled plan diversity and fills them via a beam search that branches only on schema names.

- We show that LogicalBeam consistently increases the fraction of time when all gold SQLs get generated in the Top-5 choices by $1.5 - 2.5\times$ over the baselines across the board on AmbiQT.

## 2 Background and Related Work

A Text-to-SQL model takes as input a question expressed as a natural language text $\mathbf{x}$, and a database schema $\mathbf{s}$ comprising of table and column names, and outputs an SQL program $\mathbf{y}$ which can be executed against the database to answer the user's question. Figure 1 shows an example. The training data for the task comprises (text, schema, SQL) triplets spanning multiple distinct databases.

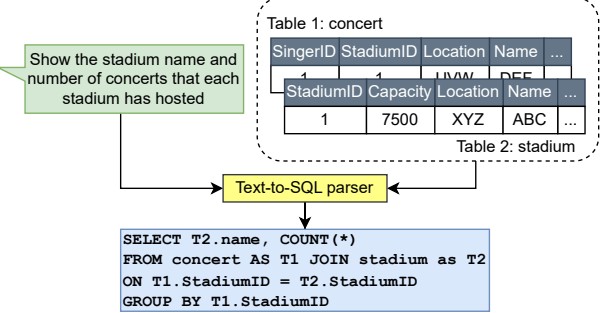

Figure 1: A Text-to-SQL system converts a user question to an SQL query, conditioned on the database schema and/or content.

**Benchmarks.** Popular benchmarks for the Text-to-SQL task are WikiSQL (Zhong et al., 2018) and SPIDER (Yu et al., 2018). A few others have been proposed recently to capture real-world scenarios, such as KaggleDBQA (Lee et al., 2021), SEDE (Hazoom et al., 2021), and EHRSQL (Lee et al., 2022). They all attach one SQL per text, though some of them mention the problem of ambiguity in real-world datasets. Dr. SPIDER (Chang et al., 2023), designed to test the robustness of existing models, perturbs either the text or schema of SPIDER but still assigns one SQL per text.

**Ambiguity in SQL** Although ambiguity has been studied in other fields of NLP (Pilault et al., 2023; Li et al., 2022; Futeral et al., 2022), it has been unexplored in the context of semantic parsing. Ambiguity in SQL arising from related column names is discussed in (Wang et al., 2022), but they only consider column ambiguity. Their method of recognizing ambiguous queries depends on labeling words of the text and does not generalize to other kinds of ambiguity. To the best of our discernment, AmbiQT represents the first open benchmark for testing coverage of ambiguous alternatives.

**Diverse Decoding.** Prior work has critiqued the lack of meaningful diversity in beam-search outputs (Finkel et al., 2006; Gimpel et al., 2013; Li et al., 2016; Li and Jurafsky, 2016). In response, many fixes have been proposed. Some proposals attempt to restrict the tokens sampled, using strategies like Nucleus sampling (Holtzman et al., 2020), Truncated Sampling (Hewitt et al., 2022), and Typical Sampling (Meister et al., 2023), while some rely on Template-Based decoding (Wiseman et al., 2018; Zhang et al., 2022; Fu et al., 2023; Elgohary et al., 2020; Awasthi et al., 2022). A third approach is to generate a prefix with high diversity first, then

| Kind of ambiguity | Count | Example | | |
|---|---|---|---|---|
| | | Question Text | SQL #1 | SQL #2 |
| Column Ambiguity (C) | 1240 | List the ids of all students. | `SELECT roll_number FROM students` | `SELECT admission_number FROM students` |
| Table Ambiguity (T) | 1417 | How many singers do we have? | `SELECT COUNT(*) FROM artist` | `SELECT COUNT(*) FROM performer` |
| Join Ambiguity (J) | 288 | What are the makers and models? | `SELECT maker, model FROM model` | `SELECT t2.maker, t1.model FROM model AS t1 JOIN model_maker AS t2 ON t1.model_id = t2.model_id` |
| Precomputed Aggregates (P) | 101 | Find the average weight for each pet type. | `SELECT AVG(weight), pettype FROM pets GROUP BY pettype` | `SELECT avg_weight, pettype FROM pets_weight` |

Table 1: The AmbiQT benchmark. For each question, we list two valid SQL queries as per the schema. The schema is not shown here, but the ambiguity in it can be inferred based on the two SQL queries.

generate the rest of the sentence with lower diversity. Narayan et al. (2022) follow this recipe but focus on incorporating diverse entity orders in text summarization.

## 3 AmbiQT: A Benchmark of Ambiguous Text-to-SQL Conversion

AmbiQT is constructed so that each text query has two distinct valid SQL interpretations. Motivated by our experience working with real-life databases, we designed AmbiQT to encompass four types of ambiguity. Each entry is designed so that both alternatives have a similar relevance to the question, and a well-calibrated decoding method is expected to rank them close by in their outputs.

We create AmbiQT by modifying the SPIDER (Yu et al., 2018) dataset, and use ChatGPT (OpenAI, 2022) to aid with the creation. In each case, we modify the schema instead of the text as that provides greater control over the modification process. We explain the kinds of ambiguity in AmbiQT below and portray examples of each in Table 1.

**Column Ambiguity (C).** Unlike the SPIDER benchmark where column names usually appear verbatim in the question text (like `born state` for the column `born_state`), when users unaware of the schema pose a natural question, they introduce column ambiguity (Wang et al., 2022). For example, "*What is the capacity of O2 Arena?*" could be ambiguous if the schema has separate columns for standing and seating capacity. Likewise, a query on the number of under-nourished children is ambiguous if we have different columns for "under-weight children" and "stunted growth in children".

To simulate column ambiguity, for each text $\mathbf{x}$, schema $\mathbf{s}$, and SQL $\mathbf{y}$ in SPIDER, we prompt ChatGPT to generate two synonyms for each column

name of $\mathbf{s}$ in a one-shot manner. Appendix A furnishes more details of the prompt. We then modify $\mathbf{s}$ by replacing $c$ with two columns $c_1, c_2$, and we use $\mathbf{y}$ to generate two queries $\mathbf{y}_1, \mathbf{y}_2$ where all mentions of $c$ are replaced with $c_1$ in $\mathbf{y}_1$ and with $c_2$ in $\mathbf{y}_2$. An example appears in the first row of Table 1. We do not reuse $c$ because the SPIDER dataset often contains column names verbatim in the question, and that would violate our attempt at keeping the two options at similar relevance levels. We modify one column at a time and generate up to 3 examples from each original entry.

**Table Ambiguity (T).** Table name ambiguity is common in databases obtained by integrating multiple data sources, as in web tables (Cafarella et al., 2008; Pimplikar and Sarawagi, 2012). Here again, we prompt ChatGPT to generate two alternate names for each table. We then modify one SQL $\mathbf{y}$ to generate two candidates $\mathbf{y}_1, \mathbf{y}_2$ as shown in Table 1.

**Join Ambiguity (J).** In production databases, a logical table is often vertically partitioned across several tables for efficient clustered access (Stonebraker et al., 2019). Column names overlapping across tables leads to Join Ambiguity. Suppose we have two tables: (1) `person` with columns `id`, `name`, `email_address`, and (2) `person_details` with columns `id`, `postal_address`, `photo`. A question asking for a person's name and address is ambiguous on whether a `JOIN` with the `person_details` is necessary. We expose such ambiguity by modifying the schema as follows.

Consider a $(\mathbf{x}, \mathbf{s}, \mathbf{y})$ triplet. Suppose $\mathbf{y}$ involves selecting two or more columns $c_1, c_2, \ldots$, not necessarily in the same order, from a table $t$. Suppose further that $c_1$ is not a primary key of $t$. We create a table called $t\_c_1$ that includes just the primary

| **Question:** Show the names of high school students and their corresponding number of friends. |
|---|
| **Gold Queries** |
| 1. SELECT t2.full_name, count(*) FROM friend AS t1 JOIN highschooler AS t2 on t1.id = t2.id GROUP BY t1.id |
| 2. SELECT t2.given_name, count(*) FROM friend AS t1 JOIN highschooler AS t2 on t1.id = t2.id GROUP BY t1.id |
| **Outputs of T5-3B with Beam Search** |
| 1. SELECT t1.given_name, count(*) FROM highschooler AS t1 JOIN friend AS t2 on t1.id = t2.id GROUP BY t1.id |
| 2. SELECT t1.given_name, count(*) FROM highschooler AS t1 JOIN friend AS t2 on t1.id = t2.id GROUP BY t1.highschooler |
| 3. SELECT t1.given_name, count(*) FROM highschooler AS t1 JOIN friend AS t2 on t1.id = t2.friend_id GROUP BY t1.grade |
| 4. SELECT t1.name, count(*) FROM highschooler AS t1 JOIN friend AS t2 on t1.id = t2.id GROUP BY t1.id |
| 5. SELECT t1.giving_name, count(*) FROM highschooler AS t1 JOIN friend AS t2 on t1.id = t2.id GROUP BY t1.id |

Figure 2: Beam Search works well when targeting only one output, but leads to superficial diversity, for example via different grouping and erroneous variants of column names.

key $pk_t$ of $t$, and $c_1$. The first alternative $\mathbf{y}_1$ is $\mathbf{y}$ and the second alternative $\mathbf{y}_2$ uses a join over $t$ and $t\_c_1$, with everything else staying the same as $\mathbf{y}$.

**Precomputed Aggregates (P):.** This ambiguity is particularly common in data warehouses such as Data Commons which pre-aggregate certain variables. For instance, the "*total rice production*" of a state might refer to the column rice_production of state rather than a sum over it. Text-to-SQL models have a bias toward introducing a sum()...group-by clause every time total appears in the text. The non-aggregated alternative is usually missing in the top-$k$ options. We incorporate this ambiguity as follows.

For each $(\mathbf{x}, \mathbf{s}, \mathbf{y})$, where $\mathbf{y}$ has at least one aggregate, we construct a new table $t'$. For each aggregate $\mathcal{A}$ over column $c$ in $\mathbf{y}$, we add to $t'$ the columns $\mathcal{A}'\_c$ for all $\mathcal{A}' \in \{\text{avg}, \text{sum}, \text{min}, \text{max}\}$, and the columns grouped by in $\mathbf{y}$. For count(*) we add a column called number. We get two gold queries, the original $\mathbf{y}$ and a second with the group-by replaced by a direct SELECT on $t'$ as shown in the example in Table 1. We also support aggregates across multiple tables but skip the details here.

## 4 Are Existing Text-to-SQL systems resilient to ambiguity?

We evaluate several SOTA Text-to-SQL models and decoding algorithms on their ability to generate the alternatives of AmbiQT in their top-$k$ outputs. Descriptions of the systems compared and evaluation metrics appear in Subsection 6.2. Table 3 features the results we obtained.

For all systems, the top-5 outputs contain both outputs only for a small percentage of the instances. To investigate the reasons for their poor coverage, we manually inspected several outputs of T5-3B

and ChatGPT. A few anecdotes for each kind of ambiguity are shown in Appendix F. The reason for the failure is that Beam Search tends to produce outputs that are minor tweaks of the best hypothesis, as also corroborated by prior work (Finkel et al., 2006; Gimpel et al., 2013; Li et al., 2016; Li and Jurafsky, 2016). One example from the 'C' split of AmbiQT that illustrates this is displayed in Figure 2. Recent diversity-promoting decoding strategies like Nucleus (Holtzman et al., 2020) and Typical (Meister et al., 2023) sampling are designed for natural language and are ineffective for capturing the structural diversity that SQL variants require. These observations motivated the design of our inference algorithm, LogicalBeam.

## 5 Our method: LogicalBeam

LogicalBeam attempts to induce *meaningful* diversity, while steering clear of vacuous forms of diversity in the formatting of the SQL. We first attempt to understand the type of logical diversity required by analyzing the errors of the top-1 output of T5-3B on the SPIDER benchmark.

The mistakes of the top-1 output are cataloged in Table 2. Apart from the column selection order, which is arguably not a serious error, the top four errors are a wrong number of joins, columns, and incorrect column and table names. A large fraction of the errors involves the "skeletal structure" of the SQL, whereas vanilla Beam Search exhibits little diversity in the SQL structure. Most of its diversity is around generating alternate forms of string literals, tweaking comparison orders, or swapping the names of temporary variables (like t1 with t2).

| Error Type | Contribution (%) |
|---|---|
| Correct Output | 70.31 |
| Wrong selection order | 7.44 |
| Missing JOIN(s) | 6.09 |
| Missing column(s) | 3.48 |
| Extra JOIN(s) introduced | 2.80 |
| Incorrect column or table names | 2.51 |
| WHERE clause missing or incorrect | 1.35 |
| Extra column introduced | 1.06 |
| ORDER BY clause missing or incorrect | 0.48 |
| GROUP BY clause missing or incorrect | 0.39 |
| Wrong comparison | 0.29 |
| DISTINCT missed or introduced | 0.10 |
| Other | 3.68 |

Table 2: A catalog of errors on the SPIDER dev split, based on Exact Match (EM), corresponding to the top-1 output from a Beam Search with a beam width of 25. Most errors stem from an incorrect number of JOINs or SELECTions, with incorrect schema names being a concern as well.

These observations drove us to design a two-stage approach. In the first stage, we generate diverse SQL skeletons (templates) to capture structural diversity, and in the second we fill in the template with schema-diverse alternatives. We illustrate our approach in Figure 3.

### 5.1 Plan-based Template Generation

A template of an SQL query abstracts away the names of the tables and columns of the SQL query, string literals, and constants, so that only the structural components (SELECTs, GROUP BYs, JOINs, comparisons and so on) remain. On the `train` split of SPIDER, we convert the gold SQL to a template by a simple rule-based replacement of schema names (details in Appendix E) and use it to train a Text-to-Template model. However, the top-$k$ templates found via beam search on this model again lacked logical diversity. One example is shown by Figure 6 in Appendix D. We thus explored a more deliberate mechanism to induce diversity following these three steps:

First, we preface a template with a plan declaring the structural properties of the SQL where diversity is desired. Based on our error analysis in Table 2, we chose to induce diversity on the number of JOINs and final SELECTions. Thus, for a given input question, we output a plan followed by a template as:

`<J> joins |  selects | <TEMPLATE>`
The left yellow box in Figure 3 shows one such plan prefixed template.

Second, we counterfactually perturb the counts in the plan as follows. We generate the top-choice template $t$ without any constraints (say, with $j$ joins and $s$ selections). We then generate four diverse plans by searching in the neighborhood of the most likely predicted structure as $(j-1, s), (j+1, s), (j, s-1), (j, s+1)$. We skip invalid combinations ($j < 0$, $j > 3$, or $s \leq 0$). We also explored sampling $j, s$ based on predicted probabilities, but these were extremely skewed.

Finally, for each plan (enforced as a prefix), we use greedy decoding to generate the template. The decoding algorithm was good at generating templates as per the specified plan.

Thus, at the end of the template generation phase, we have at most five templates.

### 5.2 Template filling with Diverse Schema

Here we fill diverse column names and table names in the generated templates. We use beam search to this end but enforce adherence to the template. We track our position in the template during infilling. If the next token is expected to not be part of a table or column name, we disallow the model from exploring anything apart from the highest-scoring next token. Otherwise, we allow it to branch in the next decoding step. However, we restrict its options to a whitelist of tokens computed beforehand by enumerating the columns/tables from the schema. The pseudocode of our Restricted Infilling method is presented in Algorithm 1.

The next challenge is how to rank the SQLs from the diverse templates and select the top-5. We initially attempted to rank based on the product of probabilities of the template and in-filling steps. However, the probability distribution of the models we worked with was extremely skewed - for example, top-$p$ sampling with $p = 0.9$ produced the same template in all infillings over $70\%$ of the time. Combined with the well-known lack of calibration of neural sequence models, we found it better to simply choose the top$-2$ SQLs from each template, along with the top$-2$ from vanilla beam-search without any templates. After filtering out duplicates, the top-5 queries in the list are returned.

## 6 Experiments

We present extensive comparisons of several State-of-the-Art Text-to-SQL models and decoding methods on AmbiQT in the following sections. We then show that LogicalBeam can be helpful even

**Algorithm 1:** Pseudocode for one Beam Extension step of the Restricted Fill-In Algorithm

**Data:** Beam width $k$, current hypotheses and scores $(\mathbf{y}_1, s_1), (\mathbf{y}_2, s_2), \cdots, (\mathbf{y}_k, s_k)$, template $\mathbf{t}$, set of all column names $C$ and table names $T$
**Result:** The next set of hypotheses with scores $(\mathbf{y}'_1, s'_1), \cdots, (\mathbf{y}'_k, s'_k)$
$H \leftarrow \emptyset$;
$U \leftarrow$ getPossibleFirstToks$(C)$ $\cup$ getPossibleFirstToks$(T)$;
**for** ( $i = 1$; $i \leq k$; $i = i + 1$ ) {
    **if** $\neg$HypConformsToTemplate$(\mathbf{y}_i, \mathbf{t})$ **then**
        /* If this hypothesis violates the template, don't extend it.        */
        **continue**;
    **end**
    $ncls \leftarrow$ getNextTokClass$(\mathbf{y}_i, \mathbf{t})$;
    /* Check if we expect to start a column/table name next.        */
    **if** $ncls \in$ {column, table} **then**
        /* Allow branching, but restrict options to whitelist.        */
        $H_i \leftarrow$ getExtensionsWithScores$(\mathbf{y}_i, s_i, U)$;
    **else**
        /* Disallow branching by only choosing the top-scoring extension        */
        $H_i \leftarrow \{$getTopExtensionWithScore$(\mathbf{y}_i, s_i)\}$;
    **end**
    $H \leftarrow H \cup H_i$
}
**return** getTopKHypothesesWithScore$(H)$;

in the absence of ambiguity. We also present a detailed ablation study of LogicalBeam, and a discussion of its use-cases and shortcomings.

## 6.1 Implementation Details of LogicalBeam

Both stages of LogicalBeam are fine-tuned versions of Flan T5-3B (max length $= 512$), with an Adafactor (Shazeer and Stern, 2018) optimizer (learning rate $1e - 4$, and no decay). The models were trained for roughly 300 epochs each, with checkpoint selection based on the highest Template Match and Exact match, respectively (on the validation set, with greedy decoding). Our datasets for the models consist of one-to-one maps of each example from SPIDER, with, e.g., the SQL query replaced by the corresponding template for the Text-to-Template model. We use the Hugging-Face LogitsProcessor[2] for the Template-Infilling model, which allows us to modify logits at each decoding step. We set all the disallowed tokens' logits to $-\infty$ to implement the restricted beam search.

## 6.2 Methods Compared

We compare with the following models. All use Beam Search with a beam width of 10 unless otherwise specified. For T5-3B (one of the best-performing baselines), alternate decoding algorithms are also included in the comparison.

**ChatGPT (CGPT):.** We prompt ChatGPT for its top five choices given the question and schema in

[2]https://huggingface.co/docs/transformers/internal/generation_utils#logitsprocessor

a one-shot manner using an example outside of AmbiQT. One-shot prompting was required to get ChatGPT to adhere to the output format. More details can be found in Appendix A. We also show in Appendix B that alternate prompts tried by prior works (such as (Liu et al., 2023)) are inefficient in getting ChatGPT to cover all possibilities.

**OpenAI Codex (Codex):.** We use few-shot prompting with the code-davinci-002 version of OpenAI Codex (Chen et al., 2021). This is the most capable Codex version at the time of writing. More details are provided in Appendix A.

**RESDSQL (RSQL):.** Among approaches that do not use ChatGPT/GPT-4, RESDSQL (Li et al., 2023) is the best-performing method on SPIDER at the time of writing. We use its 3B variant (the most potent one) for comparison but turn off the NatSQL (Gan et al., 2021) representation, as it is orthogonal to our approach and can be used with it.

**T5-3B (T5-3B):.** We use the T5-3B checkpoint from the PICARD (Scholak et al., 2021b) repository that fine-tunes T5-3B on SPIDER. By default, we use Beam Search for T5-3B.

**T5-3B with Top-$k$ sampling (T5-3B-k):.** At each step of decoding, we sample from the top-50 tokens, i.e. using top-$k$ sampling with $k = 50$.

**T5-3B with Nucleus/Top-$p$ Sampling (T5-3B-p):.** At each step of decoding, we sample from the top-$p$ tokens that account for 90% of the probability mass as proposed in (Holtzman et al., 2020).

**T5-3B with Typical Sampling (T5-3B-T):.** Typ-

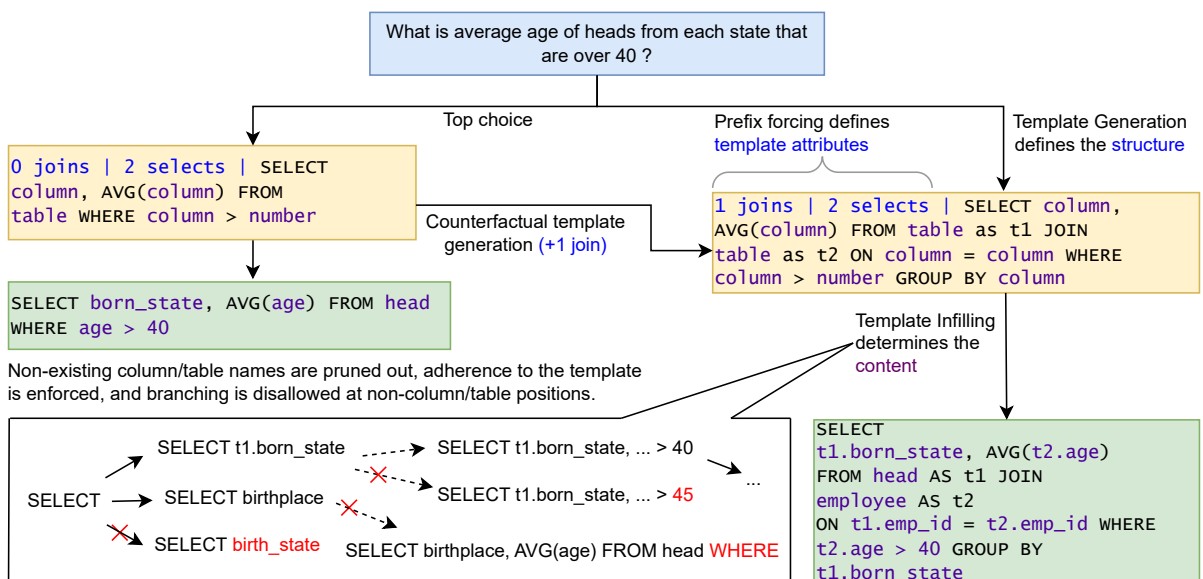

Figure 3: Our approach in its entirety. A counterfactual template generation step provides template diversity via Prefix Enforcement. Constrained infilling generates content diversity by restricting branching and enforcing template adherence.

ical Sampling (Meister et al., 2023) is another recent diverse decoding algorithm for enforcing natural diversity. This algorithm uses a parameter, `typical_p`, similar to the `top_p` of Nucleus Sampling. Following (Meister et al., 2023), we set `typical_p` to 0.9.

**Flan T5-XL (F-T5-3B):.** This is the FLAN variant of the T5-3B model, fine-tuned with the same PICARD code as the T5-3B model above.

**LogicalBeam.** For both stages we fine-tuned separate Flan T5-3B (Chung et al., 2022) models. We use a learning rate of $1 \cdot 10^{-4}$ and an effective batch size of 810 via gradient accumulation in both cases.

**Evaluation Metrics.** We present two types of accuracies (i) *EitherInTopK* - that checks if either of the gold queries feature in the top-5 outputs (ii) *BothInTopK* - that checks if both gold queries feature in the top-5. We only report the Execution Match (EXM) accuracies for each. The numbers of Exact Set Match are given in Appendix C.

### 6.3 Overall comparison on AmbiQT

We present the results of the system comparison in Table 3. We make the following observations:

- **State-of-the-art Text-to-SQL models cannot handle ambiguity:** Existing approaches, including T5-3B, ChatGPT, and RESDSQL among others, fail to cover both alternatives in Top-5 even when they perform reasonably under the *EitherInTopK* heading. Surprisingly, despite being SOTA

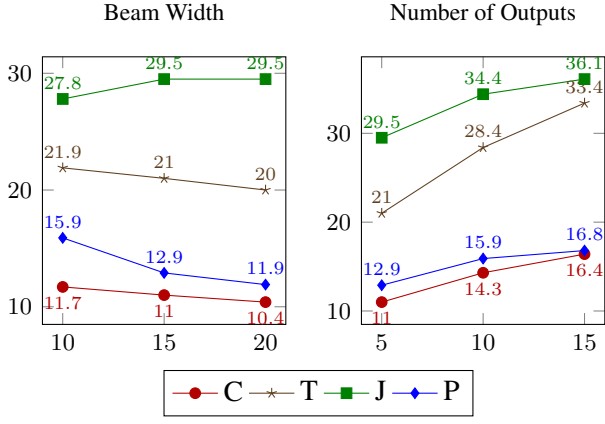

Figure 4: The coverage only increases slightly with more outputs, and *decreases* with increasing beam width. The x-axis varies the controlled hyperparameter, while the y-axis reports coverage.

on the SPIDER dataset, RESDSQL sees its coverage plummet under ambiguity. We observed that it often produced outputs that corresponded to neither of the alternatives. This behavior was also exhibited by T5-3B, by using aggregates such as `max(avg_age)`. Though outputs produced this way are syntactically correct, they do not correspond to any meaningful question.

- **Beam-search gives unhelpful token-level diversity** Although it may seem like increasing the beam width allows greater exploration and thus greater diversity, this is not the case. As Figure 4

| Kind of Ambiguity | CGPT | Codex | RSQL | F-T5-3B | T5-3B | T5-3B-k | T5-3B-p | T5-3B-T | LogicalBeam |
|---|---|---|---|---|---|---|---|---|---|
| EitherInTopK (%) | | | | | | | | | |
| C | 52.7 | 55.6 | 51.0 | 60.7 | 59.0 | 52.0 | 51.7 | 48.1 | **66.6** |
| T | 55.7 | 59.8 | 33.8 | 59.7 | 57.9 | 49.1 | 48.7 | 43.6 | **67.3** |
| J | 77.8 | 83.7 | 68.8 | 86.1 | 86.8 | 80.6 | 79.9 | 76.7 | **87.2** |
| P | 57.4 | **77.2** | 42.6 | 49.5 | 58.4 | 55.4 | 53.5 | 51.5 | 64.4 |
| BothInTopK (Coverage) (%) | | | | | | | | | |
| C | 22.7 | 10.4 | 10.8 | 8.7 | 11.7 | 3.3 | 2.5 | 0.0 | **28.0** |
| T | 37.3 | 14.3 | 6.4 | 15.1 | 21.9 | 8.0 | 7.1 | 1.1 | **42.6** |
| J | 15.6 | 43.8 | 0.0 | 22.6 | 27.8 | 2.8 | 1.7 | 0.0 | **59.4** |
| P | 8.9 | 24.7 | 8.9 | 2.9 | 15.8 | 4.0 | 3.0 | 0.0 | **24.8** |

Table 3: The results of all compared systems on AmbiQT as portrayed by Execution Match (EXM) accuracy in the Top-5 outputs. LogicalBeam usually performs the best under the EitherInTopK heading, except for Precomputed Aggregates. More importantly, LogicalBeam consistently outperforms all other systems under the BothInTopK heading. This shows the capacity of LogicalBeam to capture greater meaningful diversity in its outputs.

| Kind of Ambiguity | Single stage | Two stages | +Template Diversity | +Schema Diversity (LogicalBeam) |
|---|---|---|---|---|
| EitherInTopK (%) | | | | |
| C | 64.0 | 65.9 | 65.9 | **66.6** |
| T | 60.7 | 66.2 | 65.0 | **67.3** |
| J | 86.8 | **88.5** | 87.1 | 87.2 |
| P | 58.4 | 62.4 | 63.4 | **64.4** |
| BothInTopK (Coverage) (%) | | | | |
| C | 23.2 | 16.0 | 16.1 | **28.0** |
| T | 25.3 | 28.4 | 28.2 | **42.6** |
| J | 54.5 | 54.5 | **62.2** | 59.4 |
| P | 9.9 | 27.7 | **30.7** | 24.8 |

Table 4: The Execution Match (EXM) accuracies of the ablations on AmbiQT. Template Diversity helps with Join Ambiguity and Precomputed Aggregates, while Schema Diversity aids with Column/Table ambiguity.

shows, increasing beam width generally *reduces* coverage since the model is naturally biased towards one of the two alternatives, and a greater beam width only serves to let the model discover more vacuous variants of it. Using a larger number of outputs does not help much, as attested to by Figure 4. Even a 3× increase in the number of outputs leads only to marginal improvements, except for Table Ambiguity.

- **Recent diversity-promoting decoding algorithms fail due to skewed token distribution** SOTA diversity-promoting alternatives to Beam Search such as Nucleus and Typical sampling perform *worse* than beam search. Although these have demonstrated strong performance for tasks such as text summarization (Meister et al., 2023), they cannot promote meaningful diversity in se-

mantic parsing under ambiguity since the model produces skewed token probabilities. This causes the sampled hypotheses to be often identical as seen in the anecdotes in Figure 5 in the supplementary material.

- **LogicalBeam yields substantial accuracy in covering both ambiguous outputs**. Its BothInTopK accuracy is almost **2.5×** better in the case of Column Ambiguity than T5-3B, and **2×** in the case of Table ambiguity. It outperforms other systems by a huge margin. The accuracy under Join Ambiguity increases by **1.4×** over Codex (the next-best method) and is over **2×** better than any other method. For Precomputed Aggregates, LogicalBeam is once again the best system, and surpasses everything apart from Codex by over **1.5×**. We also convincingly beat ChatGPT and OpenAI Codex across the board on coverage.

## 6.4 Performance on Unambiguous Queries

Although our main focus is coverage under ambiguity, we also evaluate our proposal against the baseline T5-3B model on the dev split of SPIDER. We find that LogicalBeam doesn't just help the AmbiQT benchmark but also provides gains on conventional Text-to-SQL benchmarks like SPIDER where ambiguity is limited. Table 5 shows that LogicalBeam improves the top-5 Exact-Set and Execution Match accuracies on SPIDER by **2.3%** and **3.1%** over the baseline, respectively. As another example, we evaluate our method on the dev split of the challenging Kaggle DBQA (Lee et al., 2021) benchmark. We observe a drastic increase in the top-5 Exact-Set and Execution Match accuracies, from **27.1%** and **26.5%** to **35.4%** and

| Method | Top-5 Exact Set Match | Top-5 Execution Accuracy |
|---|---|---|
| SPIDER (dev split, %) | | |
| T5-3B | 76.1 | 78.2 |
| LogicalBeam | **78.4** | **81.3** |
| Kaggle DBQA (dev split, %) | | |
| T5-3B | 27.1 | 26.5 |
| LogicalBeam | **35.4** | **35.4** |

Table 5: The Exact-Set and Execution Match accuracies of LogicalBeam on two popolar Text-to-SQL datsets, SPIDER and Kaggle DBQA. Despite the datasets not exhibiting ambiguity, LogicalBeam delivers significant improvements over the T5-3B baseline.

**35.4%**, respectively. We conclude that Logical-Beam is useful across a wide range of Semantic Parsing tasks. Unlike earlier grammar-based generators like SmBoP (Rubin and Berant, 2021) that require special decoder models, our approach can work within existing LM-based models.

### 6.5 Ablation study

LogicalBeam has three design decisions: (1) Use of a two-step approach, (2) Counterfactual structural directives via plans, (3) Template-guided schema diversity. We present an ablation study where we incrementally add these changes in Table 4. The first column ("Single Stage") generates an SQL directly with a prefix for structural diversity, differing from LogicalBeam only in using a single stage. It still uses plan enforcement and branching control. We find that its coverage lags behind LogicalBeam, and by a large margin for T and P. The primary reason could be that template-guided decoding allows us to discard erroneous extensions at *each* decoding step. The second column ("Two Stages") shows a simple two-stage method where we generate a template without any counterfactual control, and use Beam Search to fill it in. This method decouples template and schema diversity, but cannot encourage either by itself. Forcing counterfactual diversity ("+Template Diversity") boosts the coverage under Join Ambiguity and Precomputed Aggregates. Finally, encouraging Schema Diversity via our Restricted Fill-In Algorithm (LogicalBeam, the last column) significantly improves coverage for Column and Table Ambiguity.

### 6.6 Discussion

LogicalBeam is general and need not be confined to the world of Semantic Parsing. For instance, the plan (prefix) could be replaced with any aspect of a code snippet that we wish to control. More generally, since the underlying mechanism only involves the model being faithful to the prefix and has no manual components, we could do the same with almost any Sequence-to-Sequence task (for example, political alignment in news summarization).

LogicalBeam consistently improves performance both under ambiguity and in the absence of it, often by drastic margins. However, we would also like to highlight one failure mode we observed, that was also exhibited by other approaches. Consider a query "`... table1 as t1 JOIN table2 as t2`". On rare occasions, we observed that an identical query with `t2` replaced by `t3` (and `t2` skipped) was also present in the choices. We believe this indicates a strong bias of the underlying model towards a particular template – so much so that it prefers this weird (`t1, t3`) combination to introducing template diversity. The problem of debiasing the model makes for exciting future work. It is not unique to Semantic Parsing, and, we believe, deserves attention in its own right.

## 7 Conclusion

In this work, we highlighted the lack of evaluation of Text-to-SQL models under ambiguity in contemporary literature. To address this, we developed AmbiQT, a novel benchmark with 3000+ challenging examples that evaluates Text-to-SQL models on four kinds of ambiguity. We demonstrated that current methods fall short of acceptable performance under ambiguity. Motivated by analyzing the errors of a T5-3B model on the SPIDER dataset, we developed a two-step approach of generating and then filling in a template. To this end, we trained a model to predict the number of `JOIN`s and selections as a plan before the template, and controlled template diversity by enforcing appropriate plans. Beam Search was modified to enforce template adherence during in-filling. Our method aligns well with intuition and greatly improves a model's coverage under ambiguity, as measured on AmbiQT. It also delivers improvements in the absence of ambiguity, on the SPIDER and Kaggle DBQA datasets. We hope our efforts inspire future work to study generation under ambiguity in more detail, both in the domain of Text-to-SQL conversion and beyond.

## Limitations

In this work, we curated a benchmark of ambiguous queries by perturbing SPIDER, an existing dataset. While we believe that our benchmark is a good measure of performance under ambiguity, real-life databases may exhibit more numerous as well as varied forms of ambiguity. In addition, AmbiQT only consists of examples with questions in English. Ambiguity may manifest differently based on the choice of natural language, and a corresponding study should make for interesting future work.

Due to the two-step approach, LogicalBeam incurs a higher number of decoding steps as compared to an end-to-end model. However, due to using a lightweight Greedy Search for the first stage, the number of decoding steps of LogicalBeam falls not much beyond the baseline. Nevertheless, finding an optimal trade-off between decoding steps and coverage remains an intriguing challenge.

At the time of writing, ChatGPT and OpenAI Codex represent the most powerful publicly available LLMs suitable for Text-to-SQL conversion and are unable to exhibit sufficient diversity under ambiguity. Future versions or models may overcome this barrier.

## Ethics Statement

As we generate AmbiQT by perturbing SPIDER (Yu et al., 2018), it does not contain any identifying information. We do not foresee any broader ethical impacts of our work.

## Acknowledgements

We thank Microsoft for sponsoring access to Azure OpenAI via the Accelerate Foundation Models Academic Research Initiative. We also extend our heartfelt gratitude to the Anonymous Reviewers who helped improve the paper with their insightful observations and suggestions.

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

## A   Computational Resources and Prompting Details

We highlight in this section the prompts we used for prompting ChatGPT, both for the synonyms of table/column names and for the Text-to-SQL conversion on AmbiQT. We also provide the prompts we used with OpenAI Codex, and furnish details of the computational resources we used. All details provided below are specified as of June 20, 2023.

### A.1   Computational Resources

All of our experiments were run on a single NVIDIA A100 GPU with 80GB of memory. We estimate the total GPU usage to have been roughly 500 GPU hours across training and inference. We

```
You are a helpful assistant that assists the user in deciding alternate names for
    ↪ their tables in an SQL database.
```

Listing 1: The directive we use for asking ChatGPT to produce table synonyms.

```
The database with database ID "[DB_ID]" currently has tables with the names [
    ↪ TABLES_STRING]. Give me two alternate names for the table "[TABLE_NAME]".
    ↪ Print your output as a python list. Do not print any additional information,
    ↪  formatting, explanation, or notes.
```

Listing 2: The prompt we use for asking ChatGPT to produce table synonyms.

further estimate the cost of utilizing ChatGPT and OpenAI Codex to be under 100$ in total.

## A.2 Synonyms (ChatGPT)

For column and table synonyms, we use one-shot prompting to indicate to ChatGPT the kind of transformation we desire.

For column synonyms, the overall directive and prompt are shown in Listings 1 and 2 respectively. The demonstrated example also follows the format of the prompt. [DB_ID] is the database ID of the database having the column, and [TABLE_NAME] is the name of the table containing it. A comma-separated list of all database table names in quotes is filled into [TABLES_STRING].

Similarly, for table synonyms, the directive and prompt are shown in Listings 3 and 4 respectively. In particular, [DB_ID] and [TABLE_NAME] are replaced with the database ID and table name. [COLUMN_NAMES] is a comma-separated list of columns of the specified table. The demonstrated example also follows this format.

We found that asking ChatGPT to structure its output as a JSON snippet saved us the trouble of sanitizing its outputs and separating it from any decoration (comments or explanation) it produced. It also made it easier to detect invalid outputs and retry.

## A.3 Text-to-SQL (ChatGPT)

We prompted ChatGPT in a one-shot manner for evaluation on our benchmark. This was necessary as our benchmark is built by modifying SPIDER (Yu et al., 2018). The queries are expected to be in a specific format in the spider dataset. In particular, the table aliases are always t1, t2, .... Further, columns are never aliased, and only unqualified JOIN is used and INNER JOINs, OUTER JOINs not used. Therefore, rather than do some ad-hoc post-correction, we showed ChatGPT one example from the original SPIDER dev set. In addition, we

asked ChatGPT to structure its output as a JSON snippet, a departure from the conventional prompt as in (Liu et al., 2023). This was motivated by our observation that ChatGPT would occasionally sneak comments or notes into its queries despite our best efforts. By asking it to produce the output in a structured (JSON) format, it was much easier to detect errors and retry.

We use the directive and prompt shown in Listings 5 and 6 respectively. The database ID and its schema go into [DB_ID] and [SCHEMA], respectively. The question is passed at the end in the placeholder [QUESTION]. Our demonstration for ChatGPT consists of using the question "*Show the stadium name and the number of concerts in each stadium*", and the output used for demonstration is shown in Listing 7.

## A.4 Text-to-SQL (OpenAI Codex)

We found that asking Codex to produce multiple SQLs in the same output did not have the desired effect, as it did not usually conform to the number of outputs or the format. Therefore, we instead prompt Codex multiple times with a temperature of 0.6 (as recommended by OpenAI to elicit creativity) and a top-$p$ of 0.7 to get its outputs. To this end, we found both zero and one-shot prompting ineffective in conveying to Codex the specific format of the output (unlike ChatGPT). In contrast, we found that few (specifically, two) shot prompting to work much better, and therefore proceeded with that alternative. Our two demonstrations as well as the query prompt follow the format of Listing 8. The output formatting is simply the SQL query string inside curly braces. The two demonstrated examples are replicated in Listing 9.

## B Alternate Prompts with ChatGPT

Before settling on our choice, we also experimented with existing prompts used by prior work (zero-shot, as opposed to our one-shot method). In

```
You are a helpful assistant that assists the user in deciding alternate names for
    ↪ their tables' columns in an SQL database.
```

Listing 3: The directive we use for asking ChatGPT to produce column synonyms.

```
The database with database ID "[DB_ID]" has a table called "[TABLE_NAME]". This
    ↪ table has columns with the following names:
[COLUMN_NAMES]
Give me two alternate names for each column. Format your output as a json snippet
    ↪ with keys corresponding to column names. Do not print any additional
    ↪ information, formatting, explanation, or notes.
```

Listing 4: The prompt we use for asking ChatGPT to produce column synonyms.

```
You are a helpful assistant that converts provided English questions to SQL queries
    ↪  with respect to a provided schema.
```

Listing 5: The directive we use while prompting ChatGPT on our benchmark.

```
The schema for a database with Database ID [DB_ID] is:
[SCHEMA]
Convert the following English question to the five most plausible SQL queries
    ↪ compatible with the above schema.
Use simply the column name for selections in simple queries. For queries with joins
    ↪ , use t1, t2, and so on as aliases for the tables, and use t1.column, t2.
    ↪ column, and so on for the column selections.
Structure your output as a JSON snippet with a single key "queries", mapping to a
    ↪ list of alternatives. Do not print any additional information, explanation,
    ↪ formatting, or notes.
Question: [QUESTION]
```

Listing 6: The prompt we use while prompting ChatGPT on our benchmark.

```
{
    "queries": [
        "select t2.name, count(*) from concert as t1 join stadium as t2 on t1.
            ↪ stadium_id = t2.stadium_id group by t1.stadium_id",
        "select t3.name, count(*) from concert as t1 join stadium as t2 on t1.
            ↪ stadium_id = t2.stadium_id join singer as t3 on t1.singer_id = t3.
            ↪ singer_id group by t1.stadium_id",
        "select t3.name, count(*) from concert as t1 join stadium as t2 on t1.
            ↪ stadium_id = t2.stadium_id join singer_in_concert as t3 on t1.
            ↪ concert_id = t3.singer_id group by t1.stadium_id",
        "select t3.name, count(*) from concert as t1 join stadium as t2 on t1.
            ↪ stadium_id = t2.stadium_id join singer_in_concert as t3 on t1.
            ↪ concert_id = t3.singer_id group by t1.stadium_id",
        "select t1.name, count(*) from stadium as t1 join concert as t2 on t1.
            ↪ stadium_id = t2.stadium_id group by t1.stadium_id"
    ]
}
```

Listing 7: The demonstrated outputs for the one-shot example with the query "*Show the stadium name and the number of concerts in each stadium*".

```
# Use the schema links to generate the SQL query for the question

[SCHEMA]

Convert the following English question to SQL queries compatible with the above
    ↪ schema.

Use simply the column name for selections in simple queries. For queries with joins
    ↪ , use t1, t2 and so on as aliases for the tables, and use t1.column, t2.
    ↪ column and so on for the column selections.

Question: [QUESTION]
```

Listing 8: The format of both the demonstrated and query examples

```
Question 1: List the official name and status of the city with the largest
    ↪ population.
Query 1: SELECT official_name, status FROM city ORDER BY population DESC LIMIT 1
Question 2: Show the stadium name and the number of concerts in each stadium.
Query 2: SELECT t1.name, count(*) FROM stadium AS t1 JOIN concert AS t2 ON t1.
    ↪ stadium_id = t2.stadium_id GROUP BY t1.stadium_id
```

Listing 9: The examples used as demonstrations for OpenAI Codex. The "Question" and "Query" indicators are just for clarity, and the formatting is as per Listing 8.

```
### Generate 5 possible sqlite SQL queries ending with ';' for the question given
    ↪ in triple backticks, with no explanation.
### Sqlite SQL tables, with their properties:
#
[DB_SCHEMA]
#
### ``` [QUESTION] ```
```

Listing 10: An alternate prompt used by prior work that we tried.

| Kind of Ambiguity | CGPT | Codex | RSQL | F-T5-3B | T5-3B | T5-3B-k | T5-3B-p | T5-3B-T | LogicalBeam |
|---|---|---|---|---|---|---|---|---|---|
| | | | | EitherInTopK (%) | | | | | |
| C | 43.4 | 51.5 | 49.4 | 58.6 | 57.6 | 52.3 | 51.9 | 48.2 | **65.8** |
| T | 41.1 | 54.3 | 30.2 | 58.9 | 56.5 | 47.6 | 47.3 | 41.9 | **66.8** |
| J | 67.4 | 82.6 | 68.1 | 87.5 | 85.4 | 82.3 | 82.3 | 79.5 | **89.2** |
| P | 51.5 | 80.2 | 57.4 | 65.4 | 78.2 | 72.3 | 74.3 | 69.3 | **81.2** |
| | | | | BothInTopK (Coverage) (%) | | | | | |
| C | 19.8 | 9.4 | 10.5 | 8.8 | 12.2 | 3.2 | 2.6 | 0.0 | **27.7** |
| T | 28.9 | 11.6 | 4.7 | 14.3 | 20.6 | 6.5 | 5.7 | 0.0 | **39.9** |
| J | 15.3 | 41.0 | 0.0 | 22.6 | 24.3 | 2.8 | 1.7 | 0.0 | **57.3** |
| P | 7.9 | **23.8** | 7.9 | 3.0 | 16.8 | 4.0 | 2.0 | 0.0 | 22.8 |

Table 6: The Exact Set Match (EM) Accuracy of the compared systems.

---

**Question**

What are the names, countries, and ages for every singer in descending order of age?

**Gold Queries**

1. SELECT name, nationality, age FROM singer ORDER BY age DESC
2. SELECT name, citizenship, age FROM singer ORDER BY age DESC

**Outputs of T5-3B with Nucleus Sampling**

1. SELECT name, nationality, age FROM singer ORDER BY age DESC
2. SELECT name, nationality, age FROM singer ORDER BY age DESC
3. SELECT name, nationality, age FROM singer ORDER BY age DESC
4. SELECT name, nationality, age FROM singer ORDER BY age DESC
5. SELECT name, nationality, age FROM singer ORDER BY age DESC

Figure 5: Nucleus Sampling shows virtually no diversity in top-5 outputs due to a highly skewed probability distribution leading to the same tokens being sampled each time.

---

| Kind of Ambiguity | Single stage | Two stages | +Template Diversity | +Schema Diversity (LogicalBeam) |
|---|---|---|---|---|
| | | | EitherInTopK (%) | |
| C | 64.3 | 64.2 | 63.6 | **65.8** |
| T | 60.0 | 64.6 | 63.2 | **66.8** |
| J | 88.9 | **90.3** | 89.2 | 89.2 |
| P | 78.2 | 80.2 | 78.2 | **81.2** |
| | | | BothInTopK (Coverage) (%) | |
| C | 23.5 | 16.0 | 16.0 | **27.7** |
| T | 20.7 | 26.8 | 26.6 | **39.9** |
| J | 53.1 | 56.6 | **64.6** | 57.3 |
| P | 7.9 | 27.7 | **29.7** | 22.8 |

Table 7: The Exact Match (EM) accuracies of the compared ablations on AmbiQT.

However, as shown in Table 8, the results with this prompting method always lag behind those obtained with our main choice.

| Kind of Ambiguity | ChatGPT (%) (Our Prompt) | ChatGPT (%) (Liu et al., 2023) |
|---|---|---|
| C | **22.7** | 11.2 |
| T | **37.3** | 15.5 |
| J | **15.6** | 0.0 |
| P | **8.9** | 6.9 |

Table 8: The results of ChatGPT with the prompt of (Liu et al., 2023) lags behind those obtained with our prompt. The better numbers are **bolded**.

particular, we tried the prompt used by (Liu et al., 2023) to evaluate ChatGPT on our benchmark with minor modifications (asking for five outputs instead of one). We showcase it in Listing 10.

Therefore, we decided to stick with our choice for the comparison in Subsection 6.3.

---

**Example Template Outputs with Beam Search**

1. SELECT column, AVG(column) FROM table GROUP BY column ORDER BY column
2. SELECT column, AVG(column) FROM table GROUP BY column ORDER BY column ASC
3. SELECT column, AVG(column) FROM table GROUP BY column ORDER BY column DESC
4. SELECT AVG(column), column FROM table GROUP BY column ORDER BY column
5. SELECT column, AVG(column) FROM table ORDER BY column

---

Figure 6: Vanilla Beam Search is inadequate to elicit meaningful template diversity. In particular, diversity in the number of JOINs or selections is lacking.

## C  Exact Set Match Accuracies for the System Comparison and Ablation Study

Here we report the Exact Match (EM) accuracies of our System Comparison and Ablation Study for both the *EitherInTopK* and *BothInTopK* modes of evaluation.

The System Comparison on AmbiQT in terms of EM, and of the various decoding algorithms, when applied to T5-3B, are shown in Table 6. We observe that Exact Set Match (EM) follows the same trend as Execution Match (EXM) under both headings, once again demonstrating the superior coverage of LogicalBeam.

The results of our Ablation Study, in turn, are shown in Table 7. The trend of EM also matches that of EXM here.

## D  Inadequacy of Conventional Decoding Algorithms

In this section, we give some anecdotes to highlight the shortcomings of conventional decoding algorithms for our purposes. The example for the case of Beam Search when used with a Text-to-SQL model was given in the main material as Figure 2. We also give here an anecdote of Nucleus Sampling in Figure 5. Strikingly, all the outputs of Nucleus Sampling are the same. This was the case for many of the examples we manually appraised. Upon further investigation, we discovered that the model produced extremely skewed probability distributions for its tokens — it was not uncommon for certain tokens to be assigned greater than 0.99 probability. This renders conventional decoding algorithms, including sampling-based methods, ineffective. Similarly, we found Beam Search (as well as sampling approaches) to be suboptimal for the case of Text-to-Template conversion, as Figure 6 exemplifies.

## E  Examples of Templates

A template is generated by abstracting away column names, table names, integer constants, and string literals from an SQL query. While these are only a small fraction of the various features of the SQL query, they represent a disproportionately large percentage of viable alternatives - for instance, a column name may be replaced by any of the numerous other ones to generate an (otherwise useless) alternative. By abstracting away these details, we avoid generating spurious alternatives by swapping these features with other ones at the template generation stage. In addition, by generating, e.g., column instead of t1.column for t1.name, we avoid trivial alias swaps. Some examples of templates for a few SQL queries are shown in Table 9, and the replacements carried out for each kind of abstraction are outlined in Table 10.

| SQL Query | Template |
|---|---|
| SELECT name FROM singer | SELECT column FROM table |
| SELECT t1.born_state, AVG(t2.age) FROM head AS t1 JOIN employee AS t2 ON t1.emp_id = t2.emp_id | SELECT column, AVG( column) FROM table AS t1 JOIN table AS t2 ON column = column |
| SELECT last_name FROM head WHERE age > 56 AND first_name = "John" | SELECT column FROM table WHERE column > number AND column = string |

Table 9: Examples of templates.

| Token type | Example | Abstraction |
|---|---|---|
| Column name | t1.name, age | column |
| Table name | singer | table |
| Number | 2,3.5 | number |
| String | "California" | string |

Table 10: The abstractions in a template.

## F Example Outputs From the Systems

We showcase example outputs from three chosen systems - our method, ChatGPT, and T5-3B on the various kinds of ambiguities of AmbiQTin Figures 7 through 10. Note that the first two outputs of our approach are from T5-3B. We observe that our approach is more consistent than the other two in incorporating all the possible queries.

# (C)olumn Synonyms

**Question**

What are the names of documents that use templates with the code BK?

**Gold Queries**

1. SELECT t2.file_name FROM templates AS t1 JOIN documents AS t2 ON t1.template_id = t2.template_id WHERE t1.template_type_code = "BK"
2. SELECT t2.record_name FROM templates AS t1 JOIN documents AS t2 ON t1.template_id = t2.template_id WHERE t1.template_type_code = "BK"

**LogicalBeam**

1. SELECT t1.file_name FROM documents AS t1 JOIN templates AS t2 ON t1.template_id = t2.template_id WHERE t2.template_type_code = "BK"
2. SELECT t1.file_name, t1.document_description FROM documents AS t1 JOIN templates AS t2 ON t1.template_id = t2.template_id WHERE t2.template_type_code = "BK"
3. SELECT t1.record_name FROM documents AS t1 JOIN templates AS t2 ON t1.template_id = t2.template_id WHERE t2.template_type_code = "BK"
4. SELECT t1.file_name FROM documents AS t1 JOIN templates AS t2 ON t1.template_id = t2.template_id WHERE t2.template_type_code = "BK" UNION SELECT t1.file_name FROM documents AS t1 JOIN templates AS t2 ON t1.template_id = t2.template_id WHERE t2.template_type_code = "BK"
5. SELECT t1.file_name FROM documents AS t1 JOIN documents AS t2 ON t1.template_id = t2.template_id WHERE t2.template_type_code = "BK"

**ChatGPT**

1. SELECT t1.record_name FROM templates AS t1 JOIN ref_template_types AS t2 ON t1.template_type_code = t2.template_type_code JOIN documents AS t3 ON t1.template_id = t3.template_id WHERE t2.template_type_code = 'BK'
2. SELECT t3.record_name FROM documents AS t3 JOIN templates AS t2 ON t3.template_id = t2.template_id JOIN ref_template_types AS t1 ON t1.template_type_code = t2.template_type_code WHERE t1.template_type_code = 'BK'
3. SELECT t1.record_name FROM templates AS t1 JOIN documents AS t2 ON t1.template_id = t2.template_id JOIN ref_template_types AS t3 ON t1.template_type_code = t3.template_type_code WHERE t3.template_type_code = 'BK'
4. SELECT t3.record_name FROM documents AS t3 JOIN templates AS t2 ON t3.template_id = t2.template_id WHERE t2.template_type_code = 'BK'
5. SELECT t1.record_name FROM templates AS t1 JOIN documents AS t2 ON t1.template_id = t2.template_id WHERE t1.template_type_code = 'BK'

**T5-3B**

1. SELECT t1.file_name FROM documents AS t1 JOIN templates AS t2 ON t1.template_id = t2.template_id WHERE t2.template_type_code = "BK"
2. SELECT t1.file_name, t1.document_description FROM documents AS t1 JOIN templates AS t2 ON t1.template_id = t2.template_id WHERE t2.template_type_code = "BK"
3. SELECT file_name FROM documents AS t1 JOIN templates AS t2 ON t1.template_id = t2.template_id WHERE t2.template_type_code = "BK"
4. SELECT t1.file_name, t1.record_name FROM documents AS t1 JOIN templates AS t2 ON t1.template_id = t2.template_id WHERE t2.template_type_code = "BK"
5. SELECT t1.document_name FROM documents AS t1 JOIN templates AS t2 ON t1.template_id = t2.template_id WHERE t2.template_type_code = "BK"

Figure 7: Example outputs from our system, ChatGPT and T5-3B on the C portion of AmbiQT

# (T)able Synonyms

**Question**

Show the name and the release year of the song by the youngest singer.

**Gold Queries**

1. SELECT song_name, song_release_year FROM artist ORDER BY age LIMIT 1
2. SELECT song_name, song_release_year FROM performer ORDER BY age LIMIT 1

**LogicalBeam**

1. SELECT song_name, song_release_year FROM artist ORDER BY age LIMIT 1
2. SELECT t2.song_name, t2.song_release_year FROM artist AS t1 JOIN song_name AS t2 ON t1.song_name = t2.song_name ORDER BY t1.age LIMIT 1
3. SELECT song_name, song_release_year FROM performer ORDER BY age LIMIT 1
4. SELECT song_name, song_release_year FROM performer ORDER BY age asc LIMIT 1
5. SELECT song_name, song_release_year FROM concert ORDER BY age LIMIT 1

**ChatGPT**

1. SELECT name, song_release_year FROM performer WHERE age = (SELECT MIN(age) FROM performer) LIMIT 1
2. SELECT name, song_release_year FROM artist WHERE age = (SELECT MIN(age) FROM artist) LIMIT 1
3. SELECT t1.name, t1.song_release_year FROM performer AS t1 JOIN singer_in_concert AS t2 ON t1.singer_id = t2.singer_id WHERE t1.age = (SELECT MIN(age) FROM performer) LIMIT 1
4. SELECT t1.name, t1.song_release_year FROM artist AS t1 JOIN singer_in_concert AS t2 ON t1.singer_id = t2.singer_id WHERE t1.age = (SELECT MIN(age) FROM artist) LIMIT 1
5. SELECT t1.name, t1.song_release_year FROM performer AS t1 JOIN singer_in_concert AS t2 ON t1.singer_id = t2.singer_id WHERE t1.age = (SELECT MIN(age) FROM (SELECT * FROM performer UNION SELECT * FROM artist)) LIMIT 1

**T5-3B**

1. SELECT song_name, song_release_year FROM artist ORDER BY age LIMIT 1
2. SELECT t2.song_name, t2.song_release_year FROM artist AS t1 JOIN song_name AS t2 ON t1.song_name = t2.song_name ORDER BY t1.age LIMIT 1
3. SELECT t2.song_name, t2.song_release_year FROM artist AS t1 JOIN song_name AS t2 ON t1.song_name = t2.song_name WHERE t1.age = (SELECT MIN(age) FROM artist)
4. SELECT song_name, song_release_year FROM artist ORDER BY age asc LIMIT 1
5. SELECT t2.song_name, t2.song_release_year FROM artist AS t1 JOIN performer AS t2 ON t1.singer_id = t2.singer_id ORDER BY t1.age LIMIT 1

Figure 8: Example outputs from our system, ChatGPT and T5-3B on the T portion of AmbiQT

# (J)oin Ambiguity

**Question**

Return the name, location, and district of all shops in descending order of number of products.

**Gold Queries**

1. SELECT name, location, district FROM shop ORDER BY number_products DESC
2. SELECT t1.name, t1.location, t2.district FROM shop AS t1 JOIN shop_district AS t2 ON t1.shop_id = t2.shop_id ORDER BY t1.number_products DESC

**LogicalBeam**

1. SELECT name, location, district FROM shop ORDER BY number_products DESC
2. SELECT name, location and district FROM shop ORDER BY number_products DESC
3. SELECT name, district, location FROM shop ORDER BY number_products DESC
4. SELECT name, district FROM shop ORDER BY number_products DESC
5. SELECT t1.name, t1.location, t2.district FROM shop AS t1 JOIN shop_district AS t2 ON t1.shop_id = t2.shop_id ORDER BY t1.number_products DESC

**ChatGPT**

1. SELECT name, location, district FROM shop JOIN shop_district ON shop.shop_id = shop_district.shop_id ORDER BY number_products DESC
2. SELECT name, location, district FROM shop_district JOIN shop ON shop.shop_id = shop_district.shop_id ORDER BY number_products DESC
3. SELECT s.name, s.location, sd.district FROM shop AS s JOIN shop_district AS sd ON s.shop_id = sd.shop_id JOIN shop AS s2 ON s2.shop_id = sd.shop_id GROUP BY s.name, sd.district, s.location ORDER BY sum(s2.number_products) DESC
4. SELECT s.name, s.location, sd.district FROM shop_district AS sd JOIN shop AS s ON s.shop_id = sd.shop_id JOIN shop AS s2 ON s2.shop_id = sd.shop_id GROUP BY s.name, sd.district, s.location ORDER BY sum(s2.number_products) DESC
5. SELECT name, location, district FROM shop_district JOIN shop ON shop.shop_id = shop_district.shop_id ORDER BY (SELECT count(*) FROM employee WHERE employee.city = shop.location) DESC

**T5-3B**

1. SELECT name, location, district FROM shop ORDER BY number_products DESC
2. SELECT name, location and district FROM shop ORDER BY number_products DESC
3. SELECT name, location, district FROM shop ORDER BY number_products DESC
4. SELECT name, location, district form shop ORDER BY number_products DESC
5. SELECT name, location, district of shop ORDER BY number_products DESC

Figure 9: Example outputs from our system, ChatGPT and T5-3B on the J portion of AmbiQT

# (P)re-Computed Aggregates

**Question**

What is the average and the maximum capacity of all stadiums?

**Gold Queries**

1. SELECT AVG(capacity), MAX(capacity) FROM stadium
2. SELECT avg_capacity, max_capacity FROM stadium_capacity

**LogicalBeam**

1. SELECT avg_capacity, max_capacity FROM stadium_capacity
2. SELECT avg_capacity (SELECT AVG(capacity) FROM stadium_capacity) and max_capacity (SELECT MAX(capacity) FROM stadium)
3. SELECT AVG(capacity), MAX(capacity) FROM stadium
4. SELECT AVG(capacity), MAX(max_capacity) FROM stadium_capacity
5. SELECT AVG(capacity), MAX(capacity), stadium_id FROM stadium_capacity

**ChatGPT**

1. SELECT AVG(capacity), MAX(capacity) FROM stadium
2. SELECT AVG(t1.capacity), MAX(t1.capacity) FROM stadium AS t1
3. SELECT AVG(t1.capacity), t1.max_capacity FROM stadium_capacity AS t1
4. SELECT AVG(t1.capacity), MAX(t2.capacity) FROM stadium AS t1 JOIN stadium AS t2 on t1.capacity <= t2.capacity group by t1.capacity
5. SELECT AVG(t2.capacity), MAX(t2.capacity) FROM concert AS t1 JOIN stadium AS t2 on t1.stadium_id = t2.stadium_id

**T5-3B**

1. SELECT avg_capacity, max_capacity FROM stadium_capacity
2. SELECT avg_capacity (SELECT AVG(capacity) FROM stadium_capacity) and max_capacity (SELECT MAX(capacity) FROM stadium)
3. SELECT avg_capacity (SELECT AVG(capacity) FROM stadium_capacity) and max_capacity (SELECT max_capacity FROM stadium_capacity) FROM stadium
4. SELECT AVG(capacity), MAX(capacity) FROM stadium_capacity
5. SELECT avg_capacity (SELECT AVG(capacity) FROM stadium_capacity) and max_capacity (SELECT MAX(capacity) FROM stadium_capacity)

Figure 10: Example outputs from our system, ChatGPT and T5-3B on the P portion of AmbiQT