# OpenReview forum: "Benchmarking and Improving Text-to-SQL Generation under Ambiguity"
_EMNLP/2023/Conference — EMNLP 2023 Main_

### Official Review · Reviewer_XzKB · 2023-07-19

**Soundness:** 4

**Excitement:**

3: Ambivalent: It has merits (e.g., it reports state-of-the-art results, the idea is nice), but there are key weaknesses (e.g., it describes incremental work), and it can significantly benefit from another round of revision. However, I won't object to accepting it if my co-reviewers champion it.

**Paper Topic And Main Contributions:**

This paper tackles the problem of ambiguity in text-to-SQL semantic parsing, where a natural language query may map to multiple possible SQL queries. The main contributions are:

- Development of AmbiQT, a new benchmark with over 3000 examples where each text maps to two valid SQL queries exhibiting different types of ambiguity (column, table, join, aggregates).
- Analysis showing that current text-to-SQL models, even large pretrained ones, do not adequately cover ambiguous alternatives in their top-k outputs.
- A new decoding algorithm, LogicalBeam, that substantially improves coverage of ambiguity by generating diverse SQL skeletons and filling them in a constrained manner.

The paper addresses an important problem in making text-to-SQL models more robust to real-world ambiguity, through both new data and methods.

**Questions For The Authors:**

- For the schema modification, was any validation done to ensure the ambiguous schemas are naturalistic?
- Were there any attempts at end-to-end training for the skeleton generation + filling steps? Could this be beneficial?
- Could the LogicalBeam approach transfer to other text-to-code generation tasks exhibiting ambiguity?
- More analysis on where LogicalBeam falls short would be useful - are there certain ambiguity patterns it still cannot handle well?
- Are there any larger implications of this work - e.g. for dialog systems that involve querying databases?

**Reasons To Accept:**

- Thoroughly motivates the problem of ambiguity in text-to-SQL, which has been lacking in prior benchmarks. AmbiQT fills this gap through a principled data collection process.
- Provides useful analysis and insights on limitations of current models in handling ambiguity.
- Proposes a novel inference algorithm that meaningfully improves coverage over strong baselines. The two-step skeleton generation and filling approach is intuitive.
- Results are comprehensive, with comparisons to multiple state-of-the-art models and ablations. Substantial gains are shown consistently.
- Well-written and easy to follow overall. The problem is clearly defined and contributions are impactful.

**Reasons To Reject:**

- While AmbiQT is a step forward, it still only covers certain predefined types of ambiguity on a modified version of an existing dataset. Testing on truly open-domain or cross-database ambiguity is an avenue for future work.
- There could be more analysis digging into the errors of LogicalBeam to understand its limitations better.
- The approach requires training separate models for the two steps, increasing training overhead. Exploring end-to-end training could be beneficial.
- The gains on SPIDER are modest, so more analysis is needed on what types of ambiguity it helps with, and what AmbiQT adds beyond SPIDER.

**Reproducibility:**

4: Could mostly reproduce the results, but there may be some variation because of sample variance or minor variations in their interpretation of the protocol or method.

**Reviewer Confidence:**

4: Quite sure. I tried to check the important points carefully. It's unlikely, though conceivable, that I missed something that should affect my ratings.

---

> ### Author Rebuttal · Authors · 2023-08-29
>
> We extend our heartfelt appreciation to the reviewer for taking the time to leave a meticulous and discerning review. We would like to respond below to some of the points raised in the review.
>
> - **AmbiQT covers only certain predefined types of ambiguity** -- While designing AmbiQT, we tried to achieve two primary goals. First, we wanted to create a benchmark representing the most critical facets of ambiguity in real life. Second, we strived to compartmentalize the types of ambiguity as rigorously as possible, as we needed to develop programmatic means to evaluate a system’s performance on AmbiQT. As ambiguity is inherently a human/natural concept, it can be tricky to straddle the boundary between exhaustive coverage and automatic evaluation. We chose the specified categories as they allowed us to address both issues. We agree that real-life workloads probably exhibit even more ambiguity – however, our benchmark covers the most prominent types, as our error catalog corroborates (Section 5). We are excited to explore other types of ambiguity in future work.
> - **Shortcomings of LogicalBeam** -- One failure mode of existing methods also exhibited by LogicalBeam was as follows. Consider a query ` … table1 as T1 JOIN table2 as T2`. On rare occasions, we observed that an identical query with `T2` replaced by `T3` (and `T2` skipped) was also present in the choices. We believe this indicates a strong bias of the underlying model towards a particular template – so much so that it prefers this weird (`T1`, `T3`) combination to introducing template diversity. The problem of debiasing the baseline model makes for exciting future work. However, the problem of a model being biased toward its top choice w.r.t other reasonable choices is not unique to Semantic Parsing. We believe that it deserves attention in its own right.
> - **Can an end-to-end model work as well?** -- We implemented an end-to-end approach that predicts the plan (JOINs and SELECTs) followed by the SQL and have compared to it in the ablation study of Section 6.4 (called “Single Stage”). We found that the two-stage approach provides significant, sometimes substantial, improvements over the direct method. While this may seem surprising, it can be explained as follows:
>   * Since there is no template to guide the decoding in the direct approach, we can prune out illegal candidates only in retrospect when it becomes clear which positions the schema names were at, i.e., possibly many decoding steps into the future.
>   * With a two-step approach, we can prune wrong extensions earlier, allowing us to explore more fruitful options instead. A back-of-the-envelope calculation: assuming that the two-step approach admits $10\%$ fewer tokens than the direct method at each decoding step. Over 64 tokens, this translates to a $(1/0.9)^{64} \sim 1000$ times more effective search.
>   * In this regard, it is possible that a direct method that could somehow anticipate the positions with schema tokens could do even better, and constitutes future work.
> - **Experiments beyond SPIDER** -- We provide additional results on the KaggleDBQA, a challenging public benchmark for text-to-SQL generation.
>
> | Method      | Top-1 Exact Match | Top-1 Execution Accuracy | Top-5 Exact Match | Top-5 Execution Accuracy |
> |-------------|------------------:|-------------------------:|------------------:|-------------------------:|
> | Beam (Baseline)        |              24.3 |                     20.3 |              27.1 |                     26.5 |
> | LogicalBeam |              24.3 |                     20.3 |              **35.4** |                     **35.4** |
>
> The “Beam” baseline here is the default  inference method of converting text to SQL while “LogicalBeam” is our method.. We find LogicalBeam offers consistent gains in the top-5 outputs increasing execution accuracy by almost 9 points. This further demonstrates that LogicalBeam, by exploring more logical diversity during inference,  is more capable of covering the correct SQL in the Top-5.
> - **Transfer to other text-to-code tasks/larger implications of the work** -- That’s a great point! Certain parts of LogicalBeam are generally beneficial. For instance, the plan of the number of JOINs and SELECTs could be replaced with any aspect of a code snippet that we wish to control. Since the underlying mechanism only involves the model being faithful to the prefix and has no manual parts, this applies to almost any sequence-to-sequence task (for example, political alignment in news summarization, or being general/specific in a question-answering task). As a more significant implication, we realized when working with LogicalBeam that Language Models (even the larger ones like ChatGPT)  often have a single thing they want to say. Despite there being multiple reasonable outputs, they prefer one strongly and continuously try to steer the decoding to that option even when the prefix aligns better with another. We believe that teaching models to acknowledge the validity of multiple outputs (through their log probabilities) is essential as their capability increases.

---

### Official Review · Reviewer_8D4B · 2023-08-04

**Typos Grammar Style And Presentation Improvements:** N/A
**Soundness:** 4

**Excitement:**

4: Strong: This paper deepens the understanding of some phenomenon or lowers the barriers to an existing research direction.

**Missing References:**

N/A

**Paper Topic And Main Contributions:**

The paper studies the natural language ambiguity problem in text-to-SQL parsing. It has two main contributions:
1. A new benchmark, AmbiQT, synthesized by rules and ChatGPT. The new benchmark is challenging to existing text-to-SQL approaches.
2. A new constrained decoding algorithm, LogicalBeam, that can improve token and structural diversity in text-to-SQL parsers.


**Questions For The Authors:**

A. Could you provide more implementation details about the text-to-template model? For example, what is its base model? Is its training data the modified Spider training set? Any checkpoint selection methods? Any intrinsic evaluations of the quality of this text-to-template model?


**Reasons To Accept:**

1. The new benchmark, AmbiQT, is a good starting point for the text-to-SQL parsing subfield to study the ambiguity problem.
2. The experiments show that LogicalBeam decoding is a solid method for improving diversity and useful for text-to-SQL parsing in general.
3. The paper is well-motivated by analysis of problems in existing text-to-SQL models and decoding methods.


**Reasons To Reject:**

While I agree with the two lexical ambiguity categories in AmbiQT, I would rather call the structural categories “diversity”, instead of  “ambiguity”. To my understanding, the synthesized parse variants for “join ambiguity” and “precomputed aggregations” are semantically equivalent to the original ones.

**Reproducibility:**

3: Could reproduce the results with some difficulty. The settings of parameters are underspecified or subjectively determined; the training/evaluation data are not widely available.

**Reviewer Confidence:**

4: Quite sure. I tried to check the important points carefully. It's unlikely, though conceivable, that I missed something that should affect my ratings.

---

> ### Author Rebuttal · Authors · 2023-08-29
>
> We extend our gratitude for your thoughtful and considerate review. We address some of the raised concerns below.
>
> - **The categories of “Join Ambiguity” and “Precomputed Aggregates” should be called “diverse” instead of “ambiguous”** -- This is a thought-provoking point! In our work, we categorize any scenario where two semantically distinct queries are ambiguous, although it would be interesting to see if future benchmarks designate a separate category for diversity.
> For the two cases above, although sometimes two columns in different tables have the same name, the two queries they give rise to are semantically different. This is because the two columns may contain different entries, and are treated as distinct by the evaluation script. We will mention this explicitly in future versions.
> - **Details of the training process** -- Both stages of LogicalBeam (template generation and infilling) are fine-tuned from Flan T5-3B (max length 512), with an Adafactor optimizer (learning rate 1e-4 with the default HuggingFace parameters, and no decay). The models were trained for roughly 300 epochs, with checkpoint selection based on the highest template match & exact match, respectively (on the validation set, Beam Width 1 with vanilla Beam Search).
> Our datasets for the two stages are one-to-one maps of each example of SPIDER, with, e.g., the SQL query replaced by the corresponding template for the text-to-template model. This replacement is done programmatically following Table 9.
> We use the HuggingFace `LogitsProcessor` for the Template-Infilling model, which allows us to modify logits at each decoding step. We set all the disallowed tokens’ logits to $-\infty$ to implement the restricted beam search.
> - **Intrinsic evaluation of the template model** -- On the dev split of SPIDER, the final checkpoint had **top-1** and **top-5** template match accuracies of **64.7%** and **80.0%**, respectively.

---

### Official Review · Reviewer_ePD9 · 2023-08-05

**Soundness:** 4

**Excitement:**

3: Ambivalent: It has merits (e.g., it reports state-of-the-art results, the idea is nice), but there are key weaknesses (e.g., it describes incremental work), and it can significantly benefit from another round of revision. However, I won't object to accepting it if my co-reviewers champion it.

**Paper Topic And Main Contributions:**

The paper tackles the challenge of ambiguity in natural language queries over real-life databases, a significant problem that existing benchmarks do not adequately capture. By developing the AmbiQT benchmark, consisting of over 3000 examples with lexical and/or structural ambiguity, the authors provide a more realistic testing ground for Text-to-SQL systems. The paper also introduces LogicalBeam, a new decoding algorithm designed to navigate the SQL logic space. The results show that LogicalBeam significantly outperforms state-of-the-art models in generating candidate SQLs and improves Exact and Execution Match Accuracies on the SPIDER dataset.

**Reasons To Accept:**

1. The development of AmbiQT is the first benchmark specifically designed to test performance under four types of ambiguity. This will likely drive further exploration and innovation in handling ambiguity, a real-world challenge often overlooked in existing benchmarks.
2. The paper's analysis of state-of-the-art methods, including prominent models like T5-3B, RESDSQL, OpenAI Codex, and ChatGPT, provides valuable insights into how they handle ambiguity. By highlighting the limitations of these models in representing ambiguity, the paper raises awareness and encourages the development of more robust solutions.
3. LogicalBeam's consistent increase in generating all gold SQLs in the Top-5 choices by 1.5-2.5 times over the baselines on AmbiQT demonstrates its effectiveness and superiority, adding a novel and effective tool to the Text-to-SQL toolkit. Its design to specifically address ambiguity opens up new avenues for research and development in this challenging aspect of language understanding.

**Reasons To Reject:**

1. The paper need to provide a thorough explanation of the LogicalBeam algorithm, including its underlying principles, design decisions, and implementation details.
2. The evaluation of LogicalBeam is confined solely to the AmbiQT benchmark without extensive comparisons to other benchmarks, thus the results might be perceived as narrowly focused and lacking generalizability.
3. While the creation of the AmbiQT benchmark is valuable, an overemphasis on this specific benchmark without a broader context or validation on other datasets could lead to questions about its applicability and relevance to other scenarios.
4. While LogicalBeam is presented as a novel solution, there have been existing work of text2sql parsing in template generation and slot filling approaches. Proper positioning and differentiation from related work would be essential to mitigate this risk.

**Reproducibility:**

3: Could reproduce the results with some difficulty. The settings of parameters are underspecified or subjectively determined; the training/evaluation data are not widely available.

**Reviewer Confidence:**

3: Pretty sure, but there's a chance I missed something. Although I have a good feel for this area in general, I did not carefully check the paper's details, e.g., the math, experimental design, or novelty.

---

> ### Author Rebuttal · Authors · 2023-08-29
>
> Thank you for the thoughtful and comprehensive insights! We would like to address some of the concerns raised below.
>
> - **Underlying principles** -- We highlight the underlying principles that motivated LogicalBeam in Section 5 (lines 303-315). Our goal was to carefully control structural diversity and lexical diversity and suppress vacuous diversity.  The type of structural diversity we chose to explore was guided by studying the typical sources of structural errors in the default beam-search on standard benchmarks (Not the new AmbiQT benchmark, to avoid bias).
> - **Design decisions** -- Our design decisions are as follows: (Lines 310-315, Lines 529-531) These decisions allowed us to realize the underlying design principles.
>   1. Use of a two-step approach: first step to generate diverse SQL structure as template, second step to fill in diverse schema elements
>   2. Diverse plan generation  achieved via conterfactually generated plans in the prefix (number of JOINs and SELECTs) upfront during template generation (Lines 335-340).
>   3. Diverse schema explorations achieved using a whitelist of tokens at schema positions during in-filling at the time of Beam Search for controlled diversity (Lines 360-371)
> - **Implementation details** -- (Sections 5.1, 5.2, 6.1, Algorithm 1 - Section F & Page 16) Both stages of LogicalBeam are fine-tuned versions of Flan T5-3B (max length 512), with an Adafactor optimizer (learning rate 1e-4 with the default HuggingFace parameters, and no decay). The models were trained for roughly 300 epochs, with checkpoint selection based on the highest template match & exact match, respectively (on the validation set, Beam Width 1 with vanilla Beam Search).
> Our datasets for the two stages are one-to-one maps of each example of SPIDER, with, e.g., the SQL query replaced by the corresponding template for the text-to-template model.
> We use the HuggingFace `LogitsProcessor` for the Template-Infilling model, which allows us to modify logits at each decoding step. We set all the disallowed tokens’ logits to $-\infty$ to implement the restricted beam search.
>
> We will make all these points even clearer in future versions, and add more details in the appendix.
> - **Evaluation on other benchmarks/validation in a broader context** -- At the time of writing, the most prominent benchmark for text-to-SQL conversion was SPIDER. In the absence of ambiguity, our approach shows a gain of 3.1% (Execution Match) on SPIDER (section 6.3) compared to  the baseline beam-search inference.
>
> We provide additional results on the KaggleDBQA -- another challenging  public benchmark for text-to-SQL generation.
>
> | Method      | Top-1 Exact Match | Top-1 Execution Accuracy | Top-5 Exact Match | Top-5 Execution Accuracy |
> |-------------|------------------:|-------------------------:|------------------:|-------------------------:|
> | Beam (Baseline)        |              24.3 |                     20.3 |              27.1 |                     26.5 |
> | LogicalBeam |              24.3 |                     20.3 |              **35.4** |                     **35.4** |
>
> The “Beam” baseline here is the default  inference method of converting text to SQL while “LogicalBeam” is our method. We find LogicalBeam offers consistent gains in the top-5 outputs increasing execution accuracy by almost 9 points. This further demonstrates that LogicalBeam, by exploring more logical diversity during inference,  is more capable of covering the correct SQL in the Top-5.
> - **Positioning w.r.t other template generation/slot-filling approaches** -- While prior approaches have utilized template-filling approaches to Semantic Parsing, we are, to our knowledge, the first to explicitly utilize this decoupling to encourage targeted types of diversity (such as that of templates via plan enforcement). We compare to one such method, RESDSQL, in section 6.2 and show significant improvements across the board. Further, as our ablation study in section 6.4 shows, using a two-step approach decouples the types of diversity, but taking advantage of this requires plan enforcement. In future versions, we will make this more explicit by adding a separate heading for template-based approaches in the Related Works section.

---

### Meta-Review · Area_Chair_DC5G · 2023-09-15

**Recommendation:** 5

**Metareview:**

In this work the authors address the ambiguity of mapping from natural language queries into SQL queries: a reasonable case in which the NL utterance can have more than one interpretation in terms of SQL queries. To study this phenomenon and measure to which extent current NL-to-SQL models can generate all viable interpretations within the top-k candidates, the authors create the new AmbiQT dataset in which two valid SQLs are written for each NL query.

They proceed to show that, due to the token-level diversity of the beam search in SotA models,   these models fail when the two valid SQLs exhibit mutual structural diversity. To address this, the authors propose a meaningful a structural decoding algorithm that nagivates the space of SQL syntax, dubbed LogicalBeam. LogicalBeam is shown to perform comparably or better to SotA on standard NL-to-SQL evaluation datasets (on SPIDER, reported in the submission; and KaggleDBQA, additionally reported in the rebuttal) and drastically better on the new AmbiQT, i.e., in the setting for which it was primarily designed.

All of the reviewers identify the new dataset and the algorithm as valuable contribution. The few concerns, primarily about the performance of LogicalBeam on the non-ambiguous NL-to-SQL datasets have been successfully resolved in the author-reviewer discussion.

---

### Decision · Program_Chairs · 2023-10-07

**Decision:**

Accept-Main

**Comment:**

In this work the authors address the ambiguity of mapping from natural language queries into SQL queries: a reasonable case in which the NL utterance can have more than one interpretation in terms of SQL queries. To study this phenomenon and measure to which extent current NL-to-SQL models can generate all viable interpretations within the top-k candidates, the authors create the new AmbiQT dataset in which two valid SQLs are written for each NL query.

They proceed to show that, due to the token-level diversity of the beam search in SotA models,   these models fail when the two valid SQLs exhibit mutual structural diversity. To address this, the authors propose a meaningful a structural decoding algorithm that nagivates the space of SQL syntax, dubbed LogicalBeam. LogicalBeam is shown to perform comparably or better to SotA on standard NL-to-SQL evaluation datasets (on SPIDER, reported in the submission; and KaggleDBQA, additionally reported in the rebuttal) and drastically better on the new AmbiQT, i.e., in the setting for which it was primarily designed.

All of the reviewers identify the new dataset and the algorithm as valuable contribution. The few concerns, primarily about the performance of LogicalBeam on the non-ambiguous NL-to-SQL datasets have been successfully resolved in the author-reviewer discussion.